# ER Negative Breast Cancer and miRNA: There Is More to Decipher Than What the Pathologist Can See!

**DOI:** 10.3390/biomedicines11082300

**Published:** 2023-08-18

**Authors:** Ghada Chamandi, Layal El-Hajjar, Abdallah El Kurdi, Morgane Le Bras, Rihab Nasr, Jacqueline Lehmann-Che

**Affiliations:** 1Department of Anatomy, Cell Biology and Physiological Sciences, Faculty of Medicine, American University of Beirut, 11-0236 Beirut, Lebanon; gc21@aub.edu.lb (G.C.); lh85@aub.edu.lb (L.E.-H.); 2Pathophysiology of Breast Cancer Team, INSERM U976, Immunologie Humaine, Pathophysiologie, Immunothérapie (HIPI), Université Paris Cité, 75010 Paris, France; morgane.le-bras@u-paris.fr; 3Office of Basic/Translational Research and Graduate Studies, Faculty of Medicine, American University of Beirut, 11-0236 Beirut, Lebanon; 4Department of Biochemistry and Molecular Genetics, Faculty of Medicine, American University of Beirut, 11-0236 Beirut, Lebanon; ak161@aub.edu.lb

**Keywords:** breast cancer, triple negative breast cancer, molecular apocrine breast cancer, luminal androgen breast cancer, biomarker, microRNA, androgen receptor

## Abstract

Breast cancer (BC), the most prevalent cancer in women, is a heterogenous disease. Despite advancements in BC diagnosis, prognosis, and therapeutics, survival rates have drastically decreased in the metastatic setting. Therefore, BC still remains a medical challenge. The evolution of high-throughput technology has highlighted gaps in the classification system of BCs. Of particular interest is the notorious triple negative BC, which was recounted as being heterogenous itself and it overlaps with distinct subtypes, namely molecular apocrine (MA) and luminal androgen (LAR) BCs. These subtypes are, even today, still misdiagnosed and poorly treated. As such, researchers and clinicians have been looking for ways through which to refine BC classification in order to properly understand the initiation, development, progression, and the responses to the treatment of BCs. One tool is biomarkers and, specifically, microRNA (miRNA), which are highly reported as associated with BC carcinogenesis. In this review, the diverse roles of miRNA in estrogen receptor negative (ER−) and androgen receptor positive (AR+) BC are depicted. While highlighting their oncogenic and tumor suppressor functions in tumor progression, we will discuss their diagnostic, prognostic, and predictive biomarker potentials, as well as their drug sensitivity/resistance activity. The association of several miRNAs in the KEGG-reported pathways that are related to ER-BC carcinogenesis is presented. The identification and verification of accurate miRNA panels is a cornerstone for tackling BC classification setbacks, as is also the deciphering of the carcinogenesis regulators of ER − AR + BC.

## 1. Introduction

Breast cancer (BC) is depicted as the most common cancer in women, with an estimated number of 2.3 million new cases worldwide in 2020 [1]. This incidence is predicted to increase in the next 15 years due to cancer screening tests, but also because of growing risk factors like increases in excess body weight [2,3]. A recent analysis of United States (US) cancer data, by The American Society of Cancer, revealed a slow increase in BC incidence (0.5% per year) since the mid-2010s. In parallel, for 30 years, female BC mortality has decreased, and this is mainly because of earlier diagnoses and improved treatments; however, this effect has been slowing in the last few years. Thus, BC remained as among the first causes of worldwide cancer deaths in 2020 [1], with 43.2 thousand estimated deaths in the US for 2022 [4]. Although the 5-year relative survival rate of BC is 90%—constituting one of the best for prognostic cancers—late recurrences are frequent, and the survival rate decreases dramatically in the metastatic setting [5].

All of these facts highlight the fact that BC remains a medical challenge, and that it will continue to be one of the major health challenges in future years.

## 2. Breast Cancer Is a Highly Heterogeneous Disease

One of the main issues concerning BC is the high heterogeneity of the disease. Indeed, BC includes a vast array of histological and molecular subtypes [6,7] with clinical implications.

First, from a histological point of view, the large majority (70–80%) of invasive breast neoplasms occur through the infiltration of ductal carcinomas of no special type (IDC-NST), which is followed by invasive lobular carcinomas (8–15%) [8,9]. Other histologic types exist but are less common, and these include micropapillary, papillary, metaplastic, and apocrine carcinomas.

Second, at a molecular level, a variety of subtypes have been described since 2000 with high therapeutic implications. Indeed, the advances in high-throughput technologies has allowed for a better biological demonstration of the BC heterogeneity at the molecular level, raising five intrinsic subtypes, which are hierarchically clustered into luminal A, luminal B, HER2-overexpressing, basal-like, and normal-like BCs [10]. Since this first transcriptomic molecular portrait of BC, multiple histopathological and biological features have been described for the purpose of a better classification and comprehension of the breast neoplasm, and this development has continued to evolve. However, four coherent groups can recurrently be defined by gene expression profiling [11]. This could be conducted possibly by multiparameter molecular tests such as PAM50 and, as is more often the case, with surrogate approaches such as by immunohistochemistry analysis. According to the St. Gallen 2013 consensus, BC molecular subtypes are defined according to estrogen receptors (ERs), progesterone receptors (PRs), Human Epidermal Growth Factor Receptor-2 (HER2), and the proliferation marker Ki67 expression as per the following: luminal A-like (ER+/PR+, HER2−, Ki67+ < 20%); luminal B-like HER2− (ER+/PR+ < 20%, HER2−, Ki67+ ≥ 20%); luminal B-like HER2+ (ER+/PR+, HER2 overexpression); HER2 overexpressed (non-luminal (ER−, PR−); HER2 overexpression); and basal-like and/or triple-negative BC (TNBC) (ER−, PR−, HER2−) [12] (Figure 1).

The luminal A-like tumors have clear prognostic and treatment implications as they proliferate less and are endocrine sensitive, thus it confers better prognosis but have a poor response to chemotherapy [13]. Luminal B-like tumors are of a higher Ki67 expression and grade, and they have less endocrine sensitivity and poorer prognoses [13,14]. HER2 overexpression leads to bad prognosis but also to a better prediction of the response to anti-HER2 therapies, which drastically improves patient survival. However, the non-luminal HER2+ group is fast growing, more aggressive, and presents a worse prognosis than luminal groups [15]. Finally, TNBCs—which account for 20% of BCs and is defined by the absence of the three major receptors of ER, PR, and HER2—present with an aggressive behavior that have a high proliferation and the most pejorative survival rates [13]. Moreover, as defined by what they are not, TNBCs remain a highly heterogeneous subgroup that need to be better characterized.

Importantly, the accurate definition of BC is necessary for proper diagnoses and treatment strategies. The huge heterogeneity of this disease is described in the WHO tumor classification [16], which was updated in 2019 [11].

## 3. Triple Negative Breast Cancers: What Are They?

TNBCs are characterized by clinical and pathological differences, as well as by distinct molecular expression profiles that translate into distinct behaviors and responses to chemotherapy. In general, TNBCs exert higher risks of recurrence with the emergence of brain and lung metastases that occur more frequently than bone metastasis when compared to other breast subtypes. Also, TNBC metastatic diseases appears rapidly within the first 3 years after diagnosis, thus leading to bad prognosis. However, when patients do not recur during this time, the survival rate is comparable to ER+ BC. Moreover, 30–40% of TNBC patients experience a pathological complete response (pCR) after neoadjuvant chemotherapy, and this constitutes a strong surrogate marker for overall survival. Therefore, it is clear that TNBCs are not a single clinico-pathological entity, but they need a better characterization of their more homogenous entities for the optimization of treatment.

Several gene expression studies have tried to dissect this heterogeneous group [17,18,19]. Initially, Lehmann et al. described six subgroups of TNBCs: basal-like 1 (BL1), basal-like 2 (BL2), immunomodulatory (IM), mesenchymal-(M), mesenchymal stem-like (MSL) and luminal androgen receptor (LAR) [17,20]. Finally, after the removal of immunological and stromal expression signals, this classification was refined into four tumor-specific subtypes (TNBCtype-4): BL1, BL2, M, and LAR. These subtypes have clear differences in their responses to chemotherapy [20]. Nevertheless, this subtyping is not currently used in routine practice. Moreover, the LAR subtype, with luminal characteristics but androgen receptor (AR) overexpression, should certainly be considered differently. In addition, the 2019 WHO classification recognized the existence of an ER− subtype, but AR+ mammary carcinoma was categorized as a distinct type of BC [7].

## 4. Apocrine Carcinoma: Just a Histology or a Molecular Entity?

Historically, breast apocrine carcinomas were defined by their particular morphological and histological appearances, with their tumor cells possibly presenting abundant granular cytoplasm, central nuclei positions, prominent nucleoli, and gross cystic disease fluid protein-15 (GCDFP-15) positive expressions by IHC [21,22]. This particular histology is also described in rare malignant adnexal neoplasms, which most commonly arise in areas with high-apocrine-gland densities, such as the axilla.

In 2005, after the transcriptomic profiling of BC, Farmer et al. described a new subtype of BC that is characterized by a luminal expression profile without ER but AR overexpression, as well as with a morphological apocrine differentiation (which was designated by the term molecular apocrine breast cancer (MABC) [23]). Subsequently, different groups have identified the MABC in non-redundant BC datasets [24]; these MABC tumors were recurrently found to specifically overexpress the AR gene and its consecutive pathway in an ER negative context with frequent expression/amplification of HER2 [23,24]. This led to the proposal of a new BC classification by Guedj et al., who split the HER2-like subtype of Perou and Sorlie into luminal B and MABC [25]. In parallel, Lehmann et al. published the TNBC subclassification described above and defined the LAR subtype as ER−/HER2−/AR+ [17,20]. Some confusion could be induced by these different descriptions, but it can be assumed that LARs probably converge on the HER2− part of the initially described MABC [25,26] (even if this has yet to be formally proven). Altogether, these data recently contributed to the consideration of these invasive MABC/LAR carcinomas as a subgroup of its own [27], leading to its inclusion in the WHO categorization of BC. This individualization of a subtype makes sense if distinct diagnoses, prognoses, or treatments are allowed by its identification as such.

## 5. MABC/LAR: How, and Why Are they Not Identified in Routine Practice?

MABC/LAR definition is based on the gene signatures obtained by messenger RNA (mRNA) expression profiling when they are not routinely performed. Some groups, including ours, have proposed MABC mRNA signatures or surrogate immunohistochemistry (IHC) markers as they are easier to apply [24,26]. However, currently, MABC/LAR profiling is not yet systematically performed.

Nevertheless, MABC/LARs are characterized by AR overexpression, and this can be easily evaluated by pathologists. Thus, MABC/LARs are essentially characterized by AR positive IHC in the context of an absence of ER and PR expressions. AR is a member of the sex steroid hormone receptor family (like ER, PR, etc.), and it is expressed in several human tissues including the breast [28]. In the context of BC, AR is overexpressed in more than 70% of cases, so it represents the greatest largely expressed hormone receptor [29]. However, it seems clear that AR plays a different role if associated with the presence or absence of ER overexpression [30].

In the ER− MABC/LAR context, the proof of concept and clinical trials supporting the targeting of AR by anti-AR drugs has come away with modest and controversial results [31,32,33,34,35,36,37,38]. Some inconsistencies could be explained by the lack of standardized AR evaluation, which is an obstacle that constitutes a major limitation for the proper definition of the subtype. Indeed, no consensus exists for the use of specific anti-AR antibodies, protocols, and positive cut-off scores. Moreover, the comparison of AR IHC evaluation and mRNA MABC signatures has demonstrated a weak concordance between these two classification tools [26]. Finally, the identification of this subtype remains a challenge, and better means for identifying it are hence needed to refine its diagnosis, prognosis, and treatment. With respect to novel and potentially useful biomarkers, microRNA (miRNA) appears to be a promising diagnostic biomarker. Moreover, the miRNA network could also help to better define the carcinogenesis of MABC/LARs and their behavior. Accordingly, in this review, we will focus on the potential role of specific dysregulated miRNA profiles in TNBC. More interestingly (in that of the less known ER−AR+ subtypes), we will also explore new approaches in order to understand and diagnose MABC/LAR breast tumors.

## 6. Search Strategy

A search strategy was adopted for the following part of the study, and two approaches were applied. The miRNAs in TNBCs were targeted by using the PubMed medical subject heading (MeSH) database. PubMed was searched for the following: “Breast Neoplasms” [MeSH] AND “MicroRNA” [MeSH] AND biomarkers AND prognosis AND diagnosis. For miRNA-AR interaction, the following terms were searched: “MicroRNA”[MeSH Terms] AND (“receptors, androgen”[MeSH Terms] OR (“receptors”[All Fields] AND “androgen”[All Fields]) OR “androgen receptors”[All Fields] OR (“androgen”[All Fields] AND “receptors”[All Fields])) AND (“breast neoplasms”[MeSH Terms].

## 7. microRNA

miRNAs are small non-coding RNAs of about 18–25 nucleotides in length. Most of these miRNAs bind to the 3′ untranslated regions of target mRNAs, thus regulating gene expression at the post-transcriptional level and leading to mRNA cleavage, translational suppression, or deadenylation [39,40,41]. In humans, it is estimated that almost a third of mRNAs are controlled by miRNAs. In fact, this is a complex network of interactions where one miRNA may bind to as much as 200 targets, and a single gene can be regulated by various miRNAs [42,43]. Rarely does a miRNA activate mRNA translation and elevate target protein levels [44]. The miRNA-mediated regulation of gene expression was highlighted by a number of studies that revealed that miRNAs play a pivotal role in physiological and pathological processes [45,46]. miRNA dysregulation is implicated in a number of diseases, including cancer [46,47,48,49,50,51]. miRNAs are associated with cancers that are generally referred to as either oncomiRs (which are highly expressed often and can promote tumor development by the targeting of tumor suppressor genes) or tumor suppressive miRNA (which are often downregulated and inhibit cancer by regulating oncogenes [52]). Some cancer-associated miRNAs are known as context-dependent miRNAs. This is highly attributed to the fact that they can act in a tissue-specific manner so that single miRNAs can have either oncogenic or tumor suppressive roles in different cancers. Collectively, a surfeit of studies has reported alterations in miRNA expression in different types of cancers. Of particular interest, some miRNAs are related to cancer development, progression, and the response of the tumor to therapy [53,54,55]. Moreover, miRNAs can be secreted into body fluids and are referred to as circulating miRNAs [56]. They are highly stable and exist as free miRNA, or are released in exosomes [57,58]. The underlying mechanism of the relationship between tissue and circulating miRNA is not well known; yet, it seems that the extracellular miRNA levels reflect deregulated signaling pathways in cancer cells [59]. Finally, these small molecules, considered as one of the largest groups of gene regulators [60,61], are easily accessible, sensitive, specific, and stable; furthermore, they accordingly have a great potential to be considered as diagnostic, prognostic, and predictive biomarkers [46,49,62,63,64].

## 8. miRNA Implications in Breast Cancer

miRNA deregulation in BC was first reported in 2005 by Iorio, after which substantial evidence in research has depicted deregulated miRNA expression to be involved in BC initiation, progression, and metastasis [65,66,67,68,69,70]. Blenkiron et al., in 2007, analyzed the miRNA expression in human BCs and demonstrated distinct miRNA signatures for the different molecular BC subtypes [71,72]. The association of miRNA activity with BC biology and its behavior was further supported by the proof that miRNAs are implicated in the regulation of ER and HER2 [73]. Moreover, there is good evidence that miRNA expression differs between primary and metastatic BCs [74,75]. This consequently led researchers to consider miRNA signatures as potential biomarkers that would help to further the understanding of BC subtypes, as well as help to predict metastasis or therapeutic resistance, thus leading to prolonged patient survival [74,76,77].

The poor prognosis of TNBCs, as well as their aggressive behavior, frequent recurrence, and poor survival has provoked a great deal of studies, which investigated miRNA signatures as a tool through which to identify patients with TNBC apart from other BC subtypes, or from healthy individuals [60,78,79]. The dysregulation of certain miRNAs appears to also have a prognostic value in TNBCs [80]. Over the past few years, and with the advancement in sequencing, several studies identified miRNA changes that were associated with TNBC development and progression (detailed in Table 1).

Indeed, both tissue and circulating miRNAs are deregulated in TNBCs and are implicated with the various pathophysiological processes of initiation, development, and the progression of tumors, which may have the potential to help in the discovery of new diagnostic, prognostic, and therapeutic strategies.

In an effort to better understand how these miRNAs are having such an impact on TNBC carcinogenesis, we executed in-silico analysis to determine which pathways these miRNAs are regulating. First of all, we had to identify the predominant miRNAs in cases where they were not reported in the literature as 3p or 5p. This was conducted through the MiRBase Converter, which is embedded in the online miRNA Enrichment and Annotation Analaysis (miEAA) tools. We also checked the miRNA annotations through using the miRbase. After which, an over-representation analysis was performed for the dysregulated miRNAs by using (miEAA), as well as by selecting the Kyoto Encyclopedia of Genes and Genomes (KEGG) pathways database as a reference. Then, we manually filtered the results to include pathways that are solely associated to BC initiation, progression, and response to therapy. Also, only the significantly deregulated pathways were accounted for, whereby significance was determined based on there being a minimum of two miRNAs present in a pathway and those which had an adjusted *p*-value < 0.05 (Figure 2). Afterward, we identified the pathways that were found to be deregulated by a common set of more than 20 miRNAs (Figure 3). Out of the fifty-eight identified miRNA, twenty-one miRNA (hsa-miR-34a-5p; hsa-miR-93-5p; hsa-miR-124-3p; hsa-miR-15a-5p; hsa-miR-15b-5p; hsa-miR-16-5p; hsa-miR-195-5p; hsa-miR-145-5p; hsa-let-7e-5p; hsa-let-7b-5p; hsa-miR-301b-3p; hsa-miR-301a-3p; hsa-miR-30a-5p; hsa-miR-30c-5p; hsa-miR-9-5p; hsa-miR-210-3p; hsa-miR-19a-3p; hsa-miR-24-3p; hsa-miR-92a-3p; hsa-miR-222-3p; and hsa-miR-155-5p) were implicated in all of the pathways that are presented in Figure 3.

Our analysis reflects the complexity of miRNA interactions in TNBC carcinogenesis, i.e., where the existence of a set of signaling pathways that are reported to be implicated in TNBC hostility is indicated. Indeed, Javier Martinez et al. described epigenetic modifications as pivotal in TNBC development, as they appear to impact both oncogenes and tumor suppressor factors, which influence various molecular pathways such as WNT/β-catenin, MAPK, and PI3K-mTOR [237]. Another implication of WNT/β-catenin alongside JAK/STAT is that they regulate BC stem cell survival and thus raise the risk of TNBC relapse [238]. TNBCs’ genomic instability, metabolic plasticity, and mutation in genes (including p53 and MAPK influence signaling pathways) are associated with the immune response [239]. Also, several studies have described deregulated lipid metabolism as a contributor in cancer cell survival, and these studies also further showed that it was mediated by PPAR-α signaling pathway [240]. A major glitch in the treatment of TNBCs is reportedly chemoresistance. It is suggested that the EGFR-K-RAS-SIAH pathway activation is a major tumor driver in chemoresistant TNBC patients [241]; another pathway that is being investigated is cAMP and its anti-proliferative role [242]. Also, oxidative phosphorylation (OXPHOS) is associated with several cancers; however, TNBC patients with a higher expression of OXPHOS have been reported to have the worst outcome [243]. In addition, checkpoint inhibitor therapy holds promise, especially in the context of metastatic TNBCs where programmed death ligand 1 (PD-L1) and PD-1 pathways are being targeted by inhibitors in combination with other adopted treatments to try to alleviate patient response [244]. Finally, it is interesting to note that the ferroptosis pathway is largely represented. This type of cell death is increasingly studied in the context of cancer [245] in line with non-coding RNAs [246], as well as recently—in particular—in the ER−/AR+ BC subtype [247].

The predicted pathways in Figure 3 are not novel in terms of TNBC; yet, those pathways have also not been studied in terms of miRNA interaction. This sheds light on the importance of investigating the panels of miRNAs in the context of studying carcinogenesis pathways.

## 9. miRNA-Implications in AR+ Tumors

Recent investigations highlighted that AR expression may be regulated by a variety of miRNAs either directly or indirectly by affecting the expression of co-activators or co-repressors. The latter would shape the AR functions [248,249,250,251]. AR is a nuclear receptor made up of a single gene that is located on the X-chromosome [252,253,254]. Androgens are usually depicted as male hormones, yet they were found to also play important biological roles in female development and physiology [255]. Dehydroepiandrosterone sulphate (DHEAS), dehydroepiandrosterone (DHEA), androstenedione (A4), testosterone, and dihydrotestosterone (DHT) are kinds of androgenic hormones that are present in the blood stream [256].

First of all, a correlation between AR expression and miRNA is particularly depicted in prostate cancer (PC) [257,258]. This interaction was found to be associated with tumor initiation and development in PC. The androgen regulation of miRNAs was examined by Waltering et al. in 2011, where DHT was found to positively regulate 17 miRNAs, out of which only 4 (miR-10a, miR-141, miR-150, and miR-1225-5p) exhibited similar androgen regulation in both in vitro and in vivo studies [259]. AR activation in PC patients reduces miR-190a expression, thus enhancing tumor-free survival [250].

By contrast, the impact of AR in BC tumorigenesis remains controversial, for it was reported that women with increased levels of androgens have increased risk of BC, while it was also reported that AR expression is a favorable BC prognostic indicator (but it has to be noticed that this is mainly true in ER+ contexts [260,261,262]). The imbalance of miRNA levels in AR+ BC cells compared to AR− BC cells implies that miRNA has a crucial role in the function of AR in BCs [263]. However, studies on the miRNA–AR interactions in BCs are limited [257,258]. Some data indicate that miR-21, an oncomiR, is upregulated in hormone-dependent neoplasms including PC and BCs [264,265], and this is reported to reduce BC cell proliferation [130]. Interestingly, AR was found to repress the transcription of miR-21 expression [266]. This suggests that more has to be evaluated in this context.

Nevertheless, some studies have focused on BCs, especially ER− ones. Shi et al. performed miRNA expression profiling in ER−/AR+ BC and revealed a total of 153 differentially expressed miRNAs in AR+ compared to AR− BC. The most significantly upregulated miRNAs were miR-933 and miR-5793, and the most downregulated was miR-4792 [263]. miR-221 and 222 that are upregulated in BC and PC are considered as oncogenes where they promote proliferation. Of interest are the miRs that are repressed by AR [130]. Another miRNA that plays an essential role in ER−/AR+ cells is miR-30b, which has been reported to inhibit cell growth [267]. miR-9-5p has an inverse relationship with AR in BCs where it exerts an anti-proliferative role [268]. miR-328-3p suppression by DHT in MDA-MB-231, suppressed CD44 expression and consequently cell adhesion. Conversely, an opposite effect was obtained upon transfection with an AR antagonist, whereby the idea that miRNAs regulate BCs was emphasized [269]. miR-190a was previously reported to be implicated in BC metastasis [270]. miR-135b, a direct regulator of AR in PC cells, was shown to have a lower expression in ER+ breast tumors when compared to ER−, as well as a higher expression in AR-low BC patient samples. It also reduces proliferation in AR+ PC cells [260]. A study conducted by Guo et al. depicted that miR-520g-3p and miR-520h are both downregulated, and that they have a significant potential in AR+ TNBC diagnosis and prognosis [271]. miR-3163 that is downregulated in AR+ ER− tumors was found to have good prognostic value [272].

MABC/LARs, i.e., the scope of this review, are characterized by AR overexpression and hyperactivation. Little is known about the miRNAs associated with this subtype. This subtype has been investigated, in vitro, via BC cell lines, in which AR expression was shown to promote their growth [273]. Of interest, in the MDA-MB-453 cell line, is an MABC model, whereby the miRNA expression that was investigated by Lyu et al. in 2014 was found to reveal four upregulated miRNAs (let-7a, let-7b, let7-c, and let7-d), where let-7a decreased cell proliferation, invasion, and migration, as well as self-renewal capacities when treating cells with DHT. In addition, this process showed a better outcome in patients with invasive BCs [274,275]. AR activity is repressed indirectly by miR-let-7c [276]. Another study investigated the role of miR-30a in MDA-MB-453, after DHT treatment, and revealed that the stimulation of AR expression inhibits miR-30a and consequently suppresses cell growth [277]. In response to AR agonists, the miR-100 and miR-125 expression was significantly reduced in MDA-MB-453 BC cells, consequently leading to the increased expression of miR-100 and miR-125 target metalloprotease-13 (MMP13) [278].

A summary of the miRNAs implicated in AR+ BC and PC is summarized in Table 2.

## 10. Challenges

Despite the fact that BC is a highly investigated research topic, and that miRNAs can serve as a biomarker for BCs, the reports on MABC are not frequent, and—in most cases—not clear. MABC is often described as under TN in the literature but also as an ER− subtype with AR overexpression, yet the mention of the name itself is not stated. This also has an impact on the search for miRNA-MABC reports. Another obstacle with most of the miRNAs reported in the literature is the lack of full miRNA annotation. This requires the use of in silico programs to predict the isoforms of miRNAs, and these might not always end up in providing the isoform investigated in the literature. Moreover, miRNAs’ specificity is often questioned, since in many cases the data are unreproducible in different datasets. This could be explained by ethnic differences, age groups, or the standardization of miRNA quantification assays in all studies. In addition to this, pathway analysis is mostly dependent on algorithms and predictions. It is worthwhile to note that all the predicted actors need to be experimentally validated before clinical utility; however, this kind of analysis could be highly valuable for new hypotheses, and could promote further pathway explorations that could help with deciphering these poorly understood BCs. Furthermore, this inventory could be a starting point through which to develop new approaches for MABC/LAR BC subtypes by including the miRNA network in the picture.

## 11. Conclusions

Differential gene expression, epigenetic modification, IHC along with other current techniques in BC classification have revealed the huge heterogeneity of this disease. Therefore, understanding the different subtypes of BCs may benefit its diagnosis, prognosis, and therapeutics. This is essential in understanding poorly diagnosed and misclassified subtypes such as MABC/LARs, as well as the consequent impact on the health management of its corresponding patients. miRNAs are reported to be deregulated in various cancers, specifically in BC and in different BC subtypes (including ER−/AR+ ones). Hence, miRNAs are a highly stable and easily detectable molecule, and they may assist in a better understanding of MABC carcinogenesis. Thus, the verification of miRNA panels in MABC patients might create a distinctive definition of this subtype, and could depict an improved understanding of the signal networks driving the biology of MABCs. In addition to this, there is piling evidence of miRNA–AR interactions in development, as well as the progression of cancer that might elucidate on MABC initiation and progression. Moreover, specific miRNAs might actually serve as diagnostic or prognostic biomarkers, but more research needs to be conducted to verify the potential clinical application of these findings. Therefore, the search for ideal biomarkers necessitates the standardization of panels in different groups, and this is subject to continuous updates that are based on advances in research and molecular technology. In this context, exploring the state-of-the-art developments of miRNAs in the MABC/LAR subtype, and attempting to extract the main miRNAs of interest could shed light on this other level of complexity, as well as help to generate new hypotheses from new angles for approaching this BC subtype that is still poorly understood.

## Figures and Tables

**Figure 1 biomedicines-11-02300-f001:**
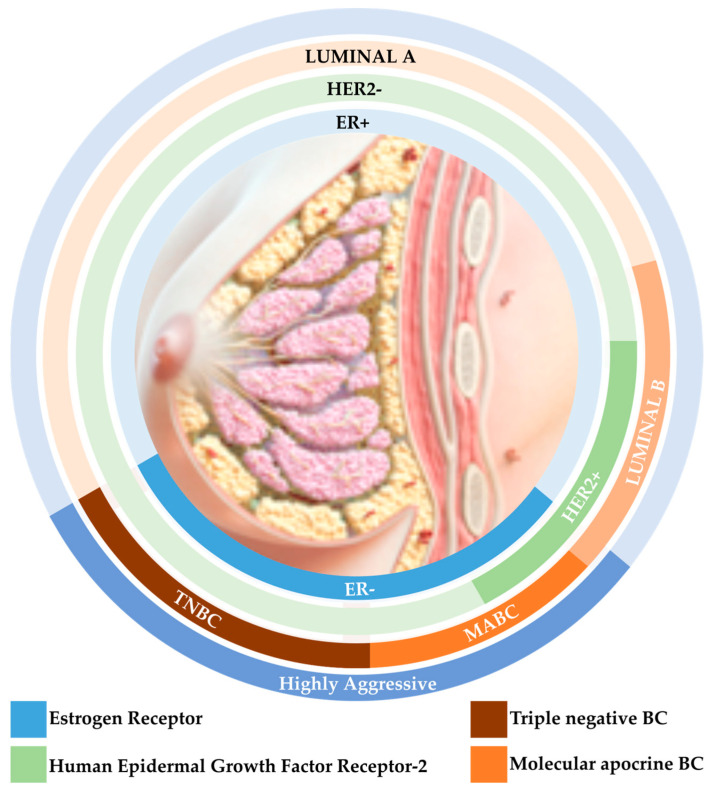
BC subtypes with respect to the receptors’ expression and correlation to aggressiveness.

**Figure 2 biomedicines-11-02300-f002:**
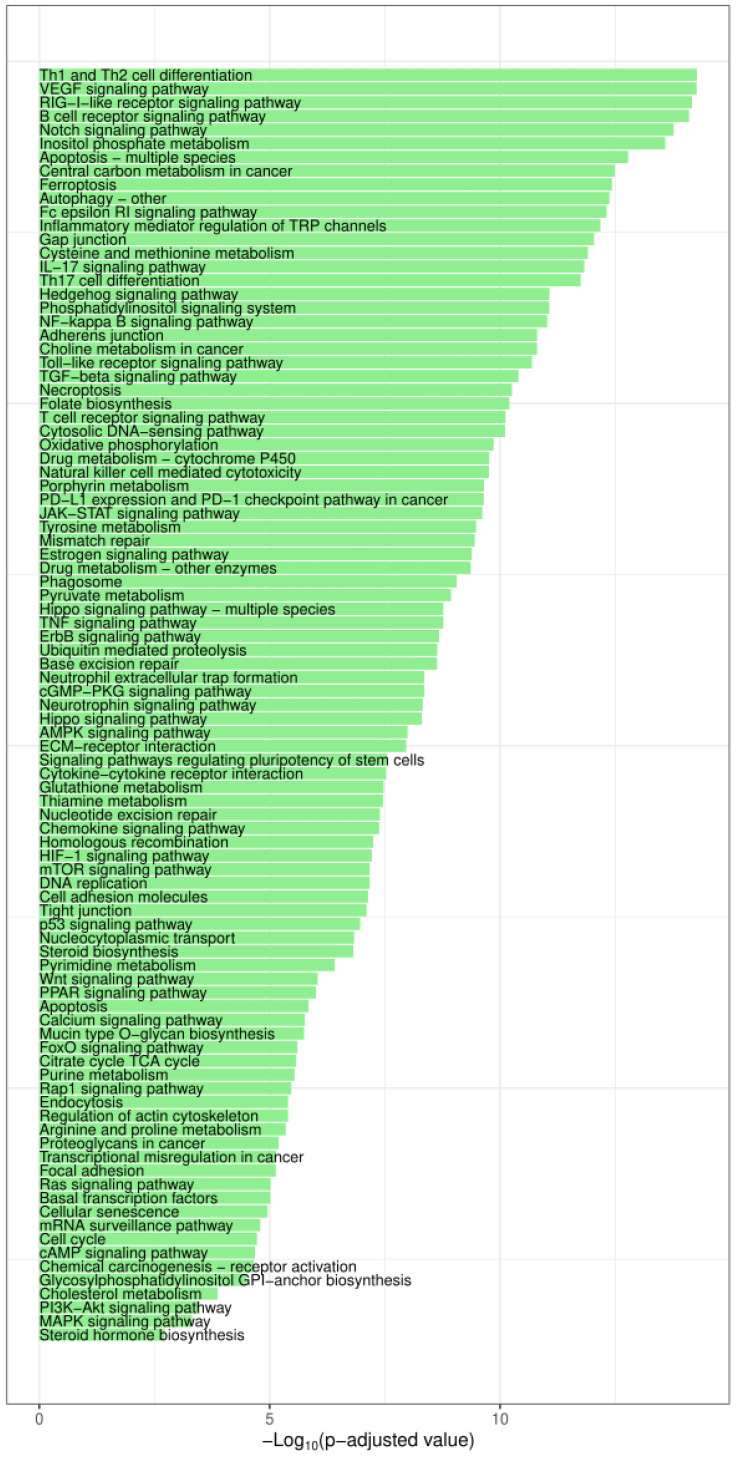
Bar plot depicting the significantly dysregulated pathways for all the dysregulated miRNAs in TNBCs, and adjusted for the decreasing *p*-values.

**Figure 3 biomedicines-11-02300-f003:**
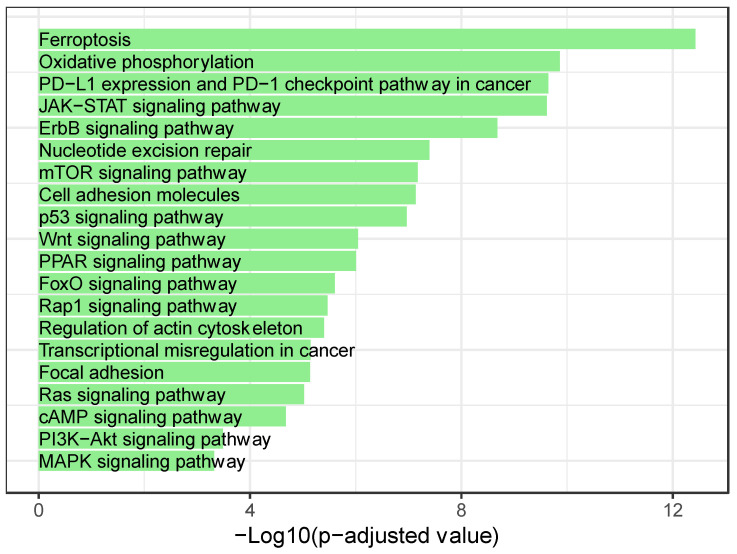
Bar plot depicting the significantly dysregulated pathways common to more than 20 miRNAs, and adjusted for the decreasing *p*-values.

**Table 1 biomedicines-11-02300-t001:** The dysregulated tissue and circulating miRNAs along with their various reported roles in TNBC carcinogenesis and their response to treatment.

miRNA Status	miRNA Annotation	Type	Role	Implications	Reference
Upregulated	miR-10b	Non-circulating	oncomiR	-Promotes proliferation, invasion, metastasis, and angiogenesis	[49,68,81,82]
	miR-181	Non-circulating	oncomiR	-Repressed by ER -Regulates the genes involved in cell growth and proliferation, including the progesterone receptor gene (a key player in estrogen signaling)	[68,83,84]
	miR-301	Non-circulating	oncomiR	-Correlates with a poor prognosis of TNBCs -Promotes the development of BCs	[85,86,87]
	miR-629-3p	Non-circulating	oncomiR	-Serves as a biomarker and a therapeutic target for lung metastasis in TNBCs	[88]
	miR-454	Non-circulating	oncomiR	-Associated with a poor prognosis and overall survival in TNBPC patients	[89]
	miR-301a	Non-circulating	oncomiR	-Correlated with a decreased overall survival and poor prognosis in TNBCs	[85,90]
	miR-182-5p	Non-circulating	oncomiR	-Promotes the proliferation and invasion of TNBCs -Associated with DNA damage repair -Correlated with cell proliferation and apoptosis	[91,92]
	miR-96-5p	Non-circulating	oncomiR	-Plays an important role in proliferation	[93]
	miR-135b	Non-circulating	oncomiR/ Suppressor	-Controls proliferation and invasion -Contributes to tumor development and progression -Worse survival in ER patients	[94,95,96]
	miR-138	Non-circulating	oncomiR	-Poor prognosis -Supports cell survival in cultures	[97]
	miR-20a-5p	Non- circulating	oncomiR	-Enhances metastasis -Implicated in apoptosis	[98,99]
	miR-455-3p	Non- circulating	oncomiR	-Improves metastasis -Increases proliferation	[100]
	miR146b-5p	Non-circulating	oncomiR	-Increases proliferation	[101]
	miR-324-5p	Non-circulating	oncomiR	-Implicated in apoptosis	[98]
	miR-939	Non-circulating	oncomiR	-Contributes to metastatic processes	[102]
	miR-362-5p	Non-circulating	oncomiR	-Facilitates proliferation and chemoresistance -Short overall survival	[103,104]
	miR-493	Non-circulating	Suppressor	-Better survival -Suppresses the invasiveness and tumorigenicity of BC cells	[105,106]
	miR-638	Non-circulating	Suppressor	-Better survival	[107]
	miR-146a	Non-circulating	Suppressor	-Better survival	[107]
	miR-182-3p	Non-circulating	Suppressor	-Reduces cell growth and activates apoptosis -Induces tumor inhibition in TNBCs	[108]
	miR-30	Non-circulating	Suppressor	-Activates p53 -Associated with good prognosis -miR-30c serves as an independent predictor in the clinical therapy of ER+ BC -Reduces cell proliferation and invasion in TNBCs	[68,109,110,111,112]
	miR-518a-3p	Non-circulating	Suppressor	-Inhibits cell migration and invasion -Better overall survival	[113]
	miR-522	Non-circulating	oncomiR	-Implicated in proliferation, invasion, and migration -High incidence of lymph node metastasis -Poor overall survival	[114]
	miR-934	Non-circulating	oncomiR	Cell proliferation	[115,116]
	miR-93-5p	Circulating	oncomiR	-Promotes chemoresistance -Acts as a diagnostic biomarker in TNBCs -Involved in TNBC metastasis and progression -Poor overall survival	[117,118,119]
	miR-105	Circulating	oncomiR	-Promotes metastasis, stemness, and chemoresistance -Poor overall survival	[118,120]
	miR-19a	Circulating	oncomiR	-Regulates anti-tumor immunity -Poor overall survival	[117,121]
	miR-19b	Circulating	oncomiR	-Promotes cell proliferation -Poor overall survival	[117,122]
	miR-22	Circulating	oncomiR	-Involved in cancer drug resistance -Promotes EMT	[117,123,124,125]
	miR-25-3p	Circulating and non-circulating	oncomiR	-Implicated in the inhibition of apoptosis -Promotes TNBC cell proliferation	[117,126]
	miR-210	Circulating and non-circulating	oncomiR	-Involved in microtubule regulation, drug efflux metabolism, and the oxidative stress response -Involved in cell proliferation, migration, and invasion -Associated with poor clinical outcomes in ER+ BC -Modulates the immune response	[68,117,127,128,129]
	miR-21	Circulating and non-circulating	oncomiR	-Promotes metastasis and proliferation -A marker of aggressiveness -Potentially prognostic in TNBC tumor stromata	[68,109,130,131,132,133,134,135,136,137,138]
	miR-19	Circulating and non-circulating	oncomiR	-Promotes EMT, migration, and invasion -Potential candidate for the diagnosis of BC when using blood samples	[139,140]
	miR-182	Circulating and non-circulating	oncomiR	-Targets the FOXO3 transcription factor expression -Promotes the macrophage activation that initiates cancer development	[141,142]
	miR-24	Circulating and non-circulating	oncomiR	-Predictor of BC relapse -Induces chemotherapy resistance -Regulates the proliferation and invasion of BC	[68,84,143,144,145]
	miR-503-3p	Circulating and non-circulating	oncomiR	-Promotes EMT	[146]
	miR-92	Circulating and non-circulating	oncomiR	-Enhances proliferation and migration	[147,148]
	miR-221/222	Circulating and non-circulating	oncomiR/ Suppressor	-Promotes EMT -Restores the expression of ER	[68,149,150,151,152]
	miR-155	Circulating and non-circulating	oncomiR/ Suppressor	-Cancer progression -Inversely correlated with the EMT in TNBCs -Associated with better clinical outcome in TNBCs -Enhances the antitumor immune response -Reverses paclitaxel resistance -A predictor of BC relapse	[53,68,109,153,154,155]
	miR-27b-3p	Circulating and non-circulating	oncomiR/ Suppressor	-A predictor of poor prognosis in invasive ductal TNBCs -Promotes tumor progression by inhibiting the peroxisome proliferator-activated receptor gamma in TNBCs	[156,157]
	miR-29a	Circulating and non-circulating	oncomiR/ Suppressor	-Promotes EMT, migration, and invasion by downregulating histone H4K20 trimethylation in TNBCs and ER+ cell lines -Decreases invasive BC cell proliferation, migration, and invasion in invasive breast cancers	[68,136,158,159]
	miR-200 family	Circulating and non-circulating	oncomiR/ Suppressor	-Promotes metastasis -Promotes EMT in aggressive cancers -Inhibits the growth and metastasis of claudin-low mammary cancers (TNBCs)	[160,161,162,163]
	miR-107	Circulating and non-circulating	oncomiR/ Suppressor	-Inhibits proliferation and migration -Associated with cell cycles, migration, invasion, revascularization, prognosis, and chemosensitivity -Improves overall survival	[98,164,165,166]
	miR-9	Circulating and non-circulating	oncomiR/Suppressor	-Associated with poor disease-free survival and distant-free survival -Enhances cell motility invasion and angiogenesis -Inhibits cell proliferation	[49,68,155,167,168]
Downregulated	miR-29c	Non-circulating	Suppressor	-Correlated with poor overall survival -Its loss is associated with the early development of TNBCs	[169]
	miR-17-5p	Non-circulating	Suppressor	-Prognostic factor for TNBCs	[170]
	miR-148a	Non-circulating	Suppressor	-Suppresses metastasis in vitro by reducing extravasation -Poor prognosis in basal and luminal B subtypes	[171]
	miR-126-5p	Non-circulating	Suppressor	-Impedes the metastasis of non-small cell lungs	[172]
	miR-1976	Non-circulating	Suppressor	-Bad overall survival -Promotes EMT	[173]
	miR-190a	Non-circulating	Suppressor	-Suppresses metastasis and angiogenesis -Correlated with a better overall survival	[96,174,175]
	miR-139-5p	Non-circulating	oncomiR	-Implicated in metastasis and chemoresistance	[176]
	miR-136-5p	Non-circulating	oncomiR	-Suppresses tumor invasion and metastasis	[96,177]
	miR-770-5p	Non-circulating	oncomiR	-Implicated in chemoresistance	[178]
	miR-4306	Non-circulating	oncomiR	-Lymph node metastasis -Poor survival -Promotes TNBC cell proliferation -Invasion and migration	[179]
	miR-196a-3p	Non-circulating	oncomiR	-Associated with lymph node metastasis -Pathological differentiation	[180]
	miR486-5p	Non-circulating	oncomiR	-Implicated in metastasis and chemoresistance	[181,182,183]
	miR-185	Non-circulating	Suppressor	-Inhibits TNBC cell proliferation	[184]
	miR-34	Non-circulating	Suppressor	-Induces apoptosis, cell cycle arrest, or senescence -Regulates cell growth, migration, invasion, angiogenesis, as well as epigenetic silencing and methylation -Promotes EMT	[49,68,109,185,186,187,188]
	miR-127	Non-circulating	Suppressor	-Suppresses proliferation, migration, and invasion -Sensitizes TNBC cells to chemotherapy	[189]
	miR-93	Non-circulating	Suppressor	-Suppresses tumor development -Enhances chemosensitivity -Mediates immunoregulation in BCs	[68,190,191,192]
	miR-124	Non-circulating	Suppressor	-Suppresses bone metastasis by repressing Interleukin-11	[193]
	miR-126	Non-circulating	Suppressor	-Associated with decreased cell proliferation -Targets the VEGF in MCF-7 cells -Inhibits the migration, invasion, and angiogenesis of TNBCs	[68,194,195,196,197]
	miR-133	Non-circulating	Suppressor	-Inhibits the growth of TNBCs	[198]
	miR-15/16	Non-circulating	Suppressor	-Inhibits cell proliferation in TNBCs -Controls angiogenesis	[199,200]
	miR-329	Non-circulating	Suppressor	-Correlates with metastasis	[201]
	miR-29a	Non-circulating	Suppressor	-Serves as a biomarker for BC diagnosis	[202]
	miR-4458	Non-circulating	Suppressor	-Regulates proliferation and apoptosis	[203]
	miR-4417	Non-circulating	Suppressor	-Prognostic biomarker for TNBCs	[204]
	miR-206	Non-circulating	oncomiR/Suppressor	-Promotes cancer progression in TNBCs and HER2+ BC by targeting neurokinin-1 receptor -Inhibits stemness and metastasis by targeting the MKL1/IL11 pathway -Suppresses EMT by targeting the TGF-β pathway in ER+ BC	[68,109,205,206,207]
	miR-31	Non-circulating	oncomiR/ Suppressor	-Correlated with poor prognosis	[208]
	miR-2117	Non- circulating	oncomiR	-Poor survival -Large tumor size	[116]
	miR-519c-3p	Non-circulating	oncomiR	-Associated with a large tumor size	[116]
	miR-873-5p	Non-circulating	Suppressor	-Promotes tumor development and metastasis	[209]
	miR-133	Non-circulating	oncomiR	-Induces proliferation and colony formation	[198]
	miR-585	Non-circulating	oncomiR	-Promotes cell proliferation, migration, and invasion -Significantly associated with poor prognosis	[210]
	miR-367	Circulating	Suppressor	-Regulates metastasis	[211]
	miR-494-3p	Circulating	oncomiR	-Implicated in immune system response	[212]
	miR-342	Circulating	Suppressor	-Biomarker for TNBCs	[168]
	miR-205	Circulating	oncomiR/ Suppressor	-Targets AR -A predictive marker of lymph node metastasis in luminal B- HER2+BC subtypes -miR-205-5p inhibits the proliferation and chemoresistance in TNBCs by targeting the HOXD9-Snail-1 axis -Expression decreases from less aggressive to more aggressive TNBCs -Inhibits proliferation and induces the EMT in TNBCs	[213,214,215,216]
	miR-199a	Circulating	oncomiR	-Affects chemosensitivity	[117,120]
	miR-195	Circulating and non-circulating	Suppressor	-Inhibits cell proliferation, glycolysis, and overall survival in ER+ BC -Differentiates metastatic BCs from the local luminal	[217,218]
	miR-205	Non-circulating	oncomiR	-Inversely associated with the tumor stage and distal metastasis of TNBCs -Poor prognosis	[219]
	Let-7 family	Circulating and non-circulating	Suppressor	-Suppresses invasion and migration -Regulates cancer stem cell properties (self-renewal, de-differentiation, and therapy resistance)	[117,220,221,222]
	miR-145	Circulating and non-circulating	Suppressor	-Suppresses metastasis and angiogenesis -Inhibits BC progression by inhibiting SOX2 -Diagnostic biomarker -Inhibits apoptosis by targeting cIAP1 (the cellular inhibitor of apoptosis)	[223,224,225,226]
	miR-335	Circulating and non-circulating	Suppressor	-Suppresses the immune escape in TNBCs -Enhances sensitivity to treatment and chemotherapy	[202,227,228,229]
	miR-128	Circulating and non-circulating	Suppressor	-Suppresses metastasis by targeting metadherin -Regulates glucose metabolism and proliferation in TNBCs	[230,231]
	miR-365	Circulating and non-circulating	Suppressor	-Anti-proliferative role -Controls invasion	[95,232]
	miR-503	Circulating and non-circulating	oncomiR/ Suppressor	-Enhances metastasis in metastatic BCs by activating the TGF-β pathway -Suppresses metastasis in ER+ BC cells -Inhibits proliferation by suppressing the CCND1 expression in BCs -Loss of miR-503 leads to chemoresistance	[233,234,235,236]

**Table 2 biomedicines-11-02300-t002:** Dysregulated tissue and the circulating miRNAs along with their various reported roles in AR+ BC and PC carcinogenesis, as well as their response to treatment.

Cancer Type	miRNA Status	miRNA Annotation	Type	Role	Implications of miRNA–AR Interaction	References
Breast cancer	Upregulated	miR-100	Non-circulating	Suppressor	-Extracellular release of MMP-13	[278,279]
		miR-125	Non-circulating	Suppressor	-Extracellular release of MMP-13	[278]
		miR-205	Non-circulating	oncomiR	-Metastasis	[213]
		miR-204	Non-circulating	Suppressor	-Promotes EMT	[280]
		miR-363	Non-circulating	oncomiR/ Suppressor	-AR induces miR-363 expression	[281]
		miR-let-7a	Non-circulating	Suppressor	-Tumor suppression, and AR induces a negative correlation between the expression of miR-let-7a and its target oncogenes of CMYC and KRAS	[274,275]
		miR-328-3p	Non-circulating	oncomiR	-Partially mediates the AR regulation of BCs	[269]
	Downregulated	miR-30a	Non-circulating	Suppressor	-Positive feedback mechanism -Suppresses cell growth	[282]
		miR-3163	Non-circulating	Suppressor	-Good prognostic role	[272]
		miR-520g-3p and miR-520h	Non-circulating	oncomiR	-Prognostic and diagnostic markers	[278]
	Differentially expressed	153 differentially expressed miRNAs in AR+ vs. AR− BC cell lines (miR-143, -4792,-145, -31, -30c, -30b-3p, 199a, and -181 downregulated in AR+ cells, while miR-933 and -5793 upregulated)	Non-circulating	oncomiR/ Suppressor	-The AR-mediated regulation of BCs is promoted by miRNAs	[263]
Prostate cancer	Upregulated	miR-17-92a	Non-circulating	oncomiR	-AR upregulates the expression of the miR-17-92a cluster	[281]
		miR-221/222	Non-circulating	oncomiR	-AR represses these miRNAs	[236]
		miR-190a	Non-circulating	oncomiR	-Contributes to tumor growth -Prognostic biomarker	[270]
	Downregulated	miR-760	Non-circulating	Suppressor	-AR downregulates miR-760, thus promoting PC growth	[283]
		miR-1205	Non-circulating	Suppressor	-Tumor suppressor	[284]
	Differentially expressed	miR-25 and miR-92b (downregulated) miR-3195, miR-3687, and miR-4417 (upregulated)	Non-circulating	oncomiR/ Suppressor	-AR upregulates the expression of these miRNAs	[285]
		miR-210-3p, miR-23c, miR-592, and miR-93-5	Circulating and non-circulating	oncomiR/ Suppressor	-Diagnostic biomarker	[286]

## Data Availability

All new data generated by in silico analysis in this study is already reported in this review.

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
