# Peer review of "ER Negative Breast Cancer and miRNA: There Is More to Decipher Than What the Pathologist Can See!"

_biomedicines, 2023, doi:10.3390/biomedicines11082300_

Round 1
Reviewer 1 Report
The manuscript by Chamandi et al. presents a comprehensive review on miRNA dysregulation in breast cancer, with a specific focus on triple-negative breast cancer (TNBC) and the MABC/LAR subtype. The following highlights several merits of the manuscript, while also identifying areas where it could be strengthened.
Merits of the manuscript:
1. Interesting and important topic: The manuscript addresses the potential implication of miRNA dysregulation in breast cancer, which is an intriguing and clinically relevant area of research. Investigating the role of miRNAs in breast cancer has the potential to provide valuable insights into disease mechanisms and identify novel therapeutic targets.
2. Good Summary of breast cancer subtype classification and challenges: The manuscript provides a good summary of breast cancer subtype classification, including the MABC and LAR subtypes. It also highlights the challenges associated with defining these special subtypes and the need for novel biomarkers in their characterization. This information is important for readers to understand the context of miRNA dysregulation in these subtypes.
3. Compilation of deregulated miRNAs in TNBC: The manuscript demonstrates substantial effort in compiling a list of miRNAs that are deregulated in TNBC. This compilation provides a valuable resource for researchers and clinicians interested in understanding the miRNA landscape in TNBC.
4. Use of in silico programs for pathway analysis: The manuscript utilizes in silico programs to analyze miRNA-regulated pathways. This approach enhances our understanding of the functional implications of miRNA dysregulation and can guide further investigations into potential therapeutic targets.
Areas for improvement:
1. While the manuscript starts by seeking potential biomarkers, it primarily compiles miRNA data from both basic cell studies and clinical samples. Many of these mRNAs might only be involved in cellular regulation, which may need more data to support them as biomarker candidates. The manuscript could benefit from acknowledging this limitation and discussing the need for further experimental validation.
2. Despite mentioning the MABC/LAR subtype as a specific focus of the paper, the amount of content dedicated to this topic seems to be limited. To strengthen the manuscript, additional information and analysis pertaining to this subtype should be included.
3. The manuscript lists a large number of deregulated miRNAs, potentially diluting the significance of individual biomarker candidates. A more critical review of the top candidates, along with a deeper discussion of their mechanistic insights and clinical relevance, would greatly enhance the manuscript.
4. The manuscript includes tables of deregulated miRNAs; however, it would be more meaningful if the regulatory molecular targets of individual miRNAs (if applicable) were also listed. This additional information would provide readers with a better understanding of the functional implications of miRNA dysregulation.
Taken together, the manuscript presents a review of miRNA dysregulation in breast cancer, with a focus on TNBC and the MABC/LAR subtype. While the manuscript covers an interesting topic and provides valuable information for further investigation, there are areas where it could be strengthened. Incorporating a critical discussion with mechanistic insights, clinical relevance, and deeper analysis of top candidates would significantly improve the manuscript. Additionally, addressing the limitations and providing further validation of potential biomarkers would enhance the overall robustness of the review.
Reviewer 2 Report
The article is a good and up-to-date collection of information on miRNA expression and function in triple negative BC and could be very useful to the BC scientific community. Figure 1 is not really needed (it can be added as a supplementary figure).
Author Response
The article is a good and up-to-date collection of information on miRNA expression and function in triple negative BC and could be very useful to the BC scientific community. Figure 1 is not really needed (it can be added as a supplementary figure).
We thank the reviewer 2 for critical reading and interest in our subject. As some discrepancies exist in the specialized literature regarding BC sub types, figure 1 is meant to illustrate how MABC is at the interface between ER-/Her2+ and ER-/Her2-, to clarify their molecular specificities. The aim was also to offer an illustration other than text and tables. However, if the reviewer feels that the figure adds nothing, we will of course agree to remove it.